# Social Media Usage and Advertising Food-Related Content: Influence on Dietary Choices of Gen Z

**DOI:** 10.3390/nu17243930

**Published:** 2025-12-16

**Authors:** Rashi Nandwani, Anu Mahajan, Vicky Wai Ki Chan, Kwok Tai Chui, Arti S. Muley, Kenneth Ka Hei Lo

**Affiliations:** 1Indian Data Capacity Accelerator, JPAL South Asia, Bangalore 560102, India; rashilnandwani@gmail.com; 2Department of Nutrition and Dietetics, Symbiosis School of Culinary Arts and Nutritional Sciences, Symbiosis International University, Pune 412115, India; anumahajan835@gmail.com; 3Department of Food Science and Nutrition, The Hong Kong Polytechnic University, Hong Kong SAR, China; waiki-vicky.chan@connect.polyu.hk; 4School of Science and Technology, Hong Kong Metropolitan University, Hong Kong SAR, China; jktchui@hkmu.edu.hk; 5Research Institute for Smart Ageing, The Hong Kong Polytechnic University, Hong Kong SAR, China

**Keywords:** social media, influencer, food porn, food blogging, food advertisements, diet culture, eating behaviour, food delivery apps

## Abstract

**Background/Objectives**: Excessive social media usage in the current times and high rates of food advertising can impact the health status of individuals by increasing food cues related to perceived hunger and, thus, dietary behaviour. This study examined the association between social media usage patterns, food-related advertising, and dietary choices among Gen Z individuals. **Methods**: A cross-sectional study was carried out amongst 314 young adults between the ages of 18 and 25 in Surat city, Gujarat. Data was collected for social media usage, the most used platforms, preferred content, and eating patterns. Anthropometric measurements (height and weight) were also recorded. Perceived hunger responses to 12 social media-based food images were assessed using a ten-point Visual Analogue Scale (VAS). Statistical analyses were performed using SPSS (version 26.0), with the significance level set at *p* < 0.05. **Results**: YouTube and Instagram were the most used social media apps. There were no significant differences observed between the BMI of participants using social media for 2 h a day and those using it 3+ hours a day. However, a significant association between the BMI of those who viewed advertisements for ready-to-eat foods (*p* = 0.004) and the BMI of those who viewed advertisements for food delivery platforms (*p* = 0.001) was seen. A significant difference between usage of Pinterest (*p* = 0.02), Instagram (*p* = 0.047), and BMI was also found. **Conclusions**: Social media marketing and food content are shaping the dietary choices of young adults, and more studies need to be conducted in Pan India to understand the reasons. Such evidence will be crucial for guiding nutrition policies, digital marketing regulations, and youth-focused awareness programmes.

## 1. Introduction

The world has undergone digital expansion. The internet has become the fourth necessity of life after food, water, and shelter, with its usage booming post-pandemic [1]. With the social isolation norms of the COVID-19 pandemic, the world moved more so to the internet to facilitate connections with a community. With 5.22 billion active users globally as of October 2024, which is 63.8% of the global population, and 256 million more users in the last year, social media usage has increased dramatically. The platform with the most active users is now YouTube, followed by Facebook and WhatsApp [1]. According to the 2024 Social Media Content Strategy Report by Sprout, the most popular social media platform among Zoomers is Instagram [2].

As of January 2024, India had 751.5 million internet users, representing 52.4% of the population, with a 2.6% increase from the previous year The most popular social platforms in India were YouTube (462.0 million users), Facebook (366.9 million users), and Instagram (362.9 million users), with YouTube and Instagram showing significant growth in terms of ad reach over the past year [2]. Young people between the age of 20–29 years, also known as Gen Z, are the most active social media users and make up about one third of the total social media users [3]. Gen Z, generally defined as individuals born between 1997 and 2012, represents the first fully digitally native cohort [4]. They differ from Millennials due to their greater dependence on smartphones and social media for information, and their decision-making and purchasing behaviour, making them especially responsive to influencer-driven marketing and visually appealing digital content. According to recent data, an average Indian spends approximately 150 min (2 h and 30 min) daily on social networking platforms in 2024 [5]. The usage varies by age group, with Gen Z (16–24 years) spending around 3 h per day on social media, while those aged 55–64 years spend about 1 h and 46 min daily [5].

Food consumption patterns are greatly impacted by digital marketing and social media influencers and celebrity endorsements, which frequently cause customers to make decisions without giving them careful thought; almost half of consumers base their purchases on influencer posts [2]. Food trends that prioritize esthetic appeal more so than nutritional content may spread quickly as a result of this impact. For example, products marketed as “no added sugar” may include substantial amounts of natural sugars or artificial sweeteners, notwithstanding the prevalence of deceptive health claims and product compositions [6]. Food advertisements (ADs) and digital marketing have significantly increased over the last few years. In 2023, the food and kindred products industry in the United States spent 3.18 billion U.S. dollars on advertising, up from roughly 2.58 billion dollars a year earlier [7]. The Centre for Disease Control also found that this persuasive food marketing increases the preference for unhealthy food choices [8]. Visual cues can be defined as different graphic elements and design styles used on landing pages, advertisements, storytelling brand videos, and other long and short forms of content that a user, who is meant to be the customer, interacts with [9]. A study found that in overweight individuals, food-related cues are an important contributor to unhealthy eating behaviours. The tempting nature of the food, owing to the way it looked and was presented, was the leading factor for unhealthy snack consumption [10].

Food marketing and ADs are well-established risk factors for obesity, especially among children. According to a Statista report, Nestle is one of the largest companies that spent about 2.4 billion dollars in AD between the years 2013 and 2020 [7]. Research has found that food influencers achieve an engagement rate of 7.38%, which is 5× engagement on their posts when compared to posts made by individual brands, companies, and restaurants [11]. Food porn is depicting food in a very tempting manner and describes indulgent food as highly tantalizing and esthetically pleasing. A study found that more than 60,000 posts are shared each day as food porn on different social media sites [12].

High screen time is an important concern today. Several observational studies have found that high screen media exposure causes an increased risk of obesity [13]. The researchers established a strong dose and response relationship between the time spent viewing television and the prevalence of being overweight in the participants [14]. Food delivery platforms across the globe have grown rapidly. The food delivery market is expected to grow by 79% in the next 5 years [15]. A report found that Zomato and Swiggy are the most widely used food delivery platforms in India [16].

Social media algorithms are designed to show users content similar to what they have previously engaged with, creating personalized feeds that can lead to filter bubbles and echo chambers [17]. Recent studies indicate that these algorithms can exacerbate mental health issues among youth, with excessive social media use linked to increased anxiety, depression, and self-harm [18]. This constant exposure to curated content and unrealistic standards can significantly impact adolescents’ self-esteem, academic performance, and overall well-being [18].

Despite extensive research on social media-driven food behaviours globally, evidence from the Indian context, particularly among Gen Z, is limited. Most existing studies focus on Western populations or broader age groups. Currently, there are limited studies that aim to understand the impact of social media usage, food content, and digital marketing ADs on the diet behaviours of young adults [19,20]. This study was based on the central research question, “How does social media usage, viewing food content, and digital marketing impact the diet behaviour and perceived hunger levels of young adults?”, with the primary objective being to determine the impact of social media marketing of food ADs and food content on the dietary choices of young adults (18–25 years). To the best of our knowledge, this is the first investigation in Gujarat (India) exploring the link between social media-based food advertising, perceived hunger, and dietary choices among Gen Z using a VAS-based craving scale and regression-based association analysis.

## 2. Materials and Methods

### 2.1. Study Design

A cross-sectional study was conducted among young adults between the age of 18–25 years residing in Surat city, Gujarat, India. Participants were recruited in person from colleges in the city using convenience sampling. Participants were recruited through convenience sampling from three colleges located in different zones of Surat city. This approach allowed the inclusion of students from varied academic backgrounds; however, the sample may not fully represent all young adults in the city. After obtaining written informed consent, participants were provided with a link to an online Google Form questionnaire, which they completed electronically. The forms were completed in the presence of the investigator.

### 2.2. Sample and Sampling Procedure

The sample size of 314 was calculated based on the prevalence of usage of social media amongst young adults (71%) [1]. The sample size was calculated using the following Cochran’s sample size formula: Z^2^ × *p*(1 − *p*)/D^2^.

Here, Z (1.96) is the Confidence interval for 95%, *p* is the prevalence of usage of social media amongst Gen Z (aged 18–25), and D (0.05) is absolute error or precision.n = Z^2^ × *p*(1 − *p*)/D^2^ = 1.96^2^ × 0.71(0.29)/0.05^2^ = 314 participants.

Because the study involved multiple behavioural and anthropometric variables, the final target sample was increased by ~10% to account for non-response and incomplete data. Ultimately, 318 participants were recruited, and 314 complete responses were included in the final analysis.

### 2.3. Ethical Considerations

The participants were informed about the study procedure, nature of the study, potential risks, and procedure. Participants who agreed to participate were asked to give informed consent. The study is approved by the University Ethics Committee (SIU/IEC/359-24.05.2022).

Inclusion Criteria: Adults aged 18–25 who regularly use Instagram, Facebook, Pinterest, and YouTube for at least 120 min daily. This threshold was selected based on recent Indian and global surveys showing that Gen Z individuals spend an average of 2–3 h per day on social media. The 120 min cutoff ensured that participants had adequate exposure to online food content, aligning with the study’s objective of examining the influence of digital advertising. However, this cutoff does not imply that individuals with lower daily usage were unimportant; rather, it was chosen to ensure that exposure to relevant stimuli (ADs and food content) was sufficiently frequent to be meaningfully assessed. Participation is open to any gender and those willing to take part in the study.

Exclusion Criteria: Young adults not using social media, individuals under 18, and those with any underlying illness, disease, or physical disability are excluded from study. Participants with self-reported chronic illnesses (e.g., endocrine disorders or renal disease), acute infections, or physical disabilities that affected appetite, mobility, or routine dietary intake were excluded. This was performed to avoid confounding the relationship between social media exposure and dietary behaviours, as certain medical conditions directly influence appetite, metabolism, or food choices. This restriction was not meant to select “healthy-only” participants but to minimize clinical conditions that could independently alter hunger cues, BMI, or responses to food-related stimuli, thereby ensuring more accurate associations.

### 2.4. Study Procedure

The anthropometric measurements, including height and weight, were directly measured by the researcher in person. These measurements were not self-reported, ensuring the accuracy and validity of the data. Height was measured using a stadiometer with participants standing barefoot, heels together, and with their heads in the Frankfurt plane. Weight was measured using a calibrated digital weighing scale, with participants wearing light clothing. Each measurement was recorded three times, and the average value was used for analysis. BMI was calculated as weight (kg)/height (m^2^) and categorized using the WHO’s adult BMI classifications [21]. These measurements were not self-reported, ensuring the accuracy and validity of the data. Next, a Google form was administered, which had questions about socio-demography, social media usage, interaction with content on social media, and 12 photos picked from social media advertisements and food content.

#### 2.4.1. Questionnaire

Data was collected using a pre-tested and self-administered questionnaire for the background information of the participants and disease history. Questions based on meal patterns were adapted from the “Global School-based Student Health Survey (GSHS) 2013” by the World Health Organization [22]. Questions for the Visual Analogue Scale were developed to measure food cue reactivity to pictorial content. This was comparable to previous studies where the VAS was used to measure perceived hunger after watching a stimulus [23]. A total of 12 pictures from Social Media Accounts were selected. A ten-point VAS was used to understand the levels of perceived hunger. The participants were asked to rate the level of perceived hunger on a scale of 1–10 (0 = no craving; 10 = very high craving) for ease of administration in an online questionnaire. The scale format was clearly explained to all participants before rating. The participants were asked to rate the level of perceived hunger on a scale of 1–10. The scale was divided into low perceived hunger levels for scores 0–3, moderate perceived hunger levels at 4–6, and 7–10 for high perceived hunger levels. Perceived hunger can be defined as eating driven by a desire for the pleasurable taste of food. The intensity of perceived hunger felt can be quantified with the help of a process that can translate sensation into action. The Visual Analogue Scale (VAS) is an effective tool for this. Twelve images were selected from Instagram food advertising posts. Items included burgers, pizzas, desserts, beverages, and Indian fast foods (all considered energy-dense, nutrient-poor foods). Sample images have been included in Appendix A.

To ensure content validity in the present study, the selected food images were reviewed by a digital marketing expert and two nutrition researchers for clarity, recognizability, and relevance to food-related content typically encountered by Gen Z individuals on social media. A pilot test was conducted prior to data collection to evaluate comprehension, image clarity, and response consistency, following which minor modifications were made.

#### 2.4.2. Measures

Social media usage, and food content and digital marketing (social media marketing) exposure.
The participants were asked to report on their use of social media, the hours spent daily on social media, the place where they use social media, and the frequency with which they use social media per day.Social media platforms where they frequently see food advertisements and content (Instagram, Facebook, YouTube, Pinterest, Snapchat, Twitter, and others).Dietary habits.
Dietary habits were assessed using a pre-tested, self-administered questionnaire adapted from the World Health Organization’s Global School-based Student Health Survey [22]. Items covered meal patterns (meals per day, breakfast consumption, snacking, and hunger between meals) and behaviours related to ordering food after viewing social-media content.Appetite responses to 12 food images were measured using a 100 mm Visual Analogue Scale (VAS), with scores summarized as mean ± SD and categorized into low, moderate, or high hunger levels.Background information.
In addition to these questions, students were asked about their disease history, gender, educational level as well as their occupation. Their anthropometric measurements of height and weight were taken.Perceived hunger scores.
The perceived hunger levels were measured using the Visual Analogue Scale.

### 2.5. Data Analysis

The data correction, entry, and coding were performed in MS Excel. The data were analyzed using SPSS (version 26.0). Descriptives summarized the sample’s characteristics. Categorical variables such as sex, occupation, and education were coded as integers following the response categories in the questionnaire. Social media usage behaviours (e.g., frequency, platform used, and type of content engaged with) were coded into ordinal categories. Visual Analogue Scale (VAS) scores ranging from one to ten for each food item were retained as continuous variables, while the total VAS score (0–120) was categorized into low, moderate, and high preference groups based on tertiles.

Potential confounders were selected based on the existing literature linking demographic factors and social media usage patterns to dietary behaviours among young adults. Variables such as age, sex, BMI category, education level, and social media usage frequency are established determinants of food choices and digital engagement and therefore were included in the adjusted regression models. These variables were not chosen based on data but on theory, following standard epidemiological guidelines for confounder selection.

Data was analyzed using SPSS (version 26.0). Descriptives summarized the sample’s characteristics. Chi-square tests assessed associations between categorical variables, and *t*-tests/ANOVA compared group differences in VAS hunger scores. Adjusted analyses were conducted using binary logistic regression (for categorical outcomes) and multiple linear regression (for continuous outcomes), with robust standard errors. Models were adjusted for age, sex, BMI category, and time spent on social media, factors which were identified a priori based on existing literature. To adjust for confounders (age, sex, BMI, occupation, and education), binary logistic and multiple linear regression models were performed with robust standard errors. The statistical significance was set at *p* < 0.05. Given the exploratory nature of the study, corrections for multiple comparisons were not applied to avoid Type II errors. However, the results are interpreted cautiously.

## 3. Results

### 3.1. Background Information

The general characteristics of the participants of the study are shown in Table 1. Out of the total of 314 participants, 146 participants were male and 168 were female, with a mean age of 19.7 (±1.50) years. Although statistical significance was observed (*p* = 0.025), the absolute difference in mean age (0.01 years) is negligible and not practically meaningful. Most of the participants, i.e., 91.4%, were higher secondary school and college students and 27 (8.4%) were working professionals. An alarming number of 126 (40.1%) participants fell under the BMI category of overweight, pre-obese, and obese according to the Asian BMI cutoff i.e., (>25 kg/m^2^). A small proportion of the participants, i.e., eight (0.5%), showed a disease history of PCOS followed by two (0.6%) participants with kidney disease and two (0.6%) with migraines.

### 3.2. Eating Patterns

Among the study population, 153 (48.7%) participants reported that they ate three meals a day and 73 (23.2%) reported that they ate more than three meals a day (Table 2). About half the participants, i.e., 157 (50%), did not consume breakfast and 87 (27.7%) consumed it sometimes. In total, 136 (43.3%) said they regularly feel hungry at mealtimes, and 109 (34.7%) said they sometimes feel hungry other than at mealtimes. The difference was not significant between boys and girls (*p* > 0.05).

### 3.3. Social Media Usage

A total of 57% of the participants spent at least 120 min/day on social media, while 16.9% of participants spent more than 180 min/day. About half of the participants, 50.6%, visited social media at least 2–5 times/day, and a total of 42.7% used social media at least 5–10 times/day.

Figure 1 presents a social media platform of choice for watching and viewing food content. The most widely used social media platforms for watching and viewing food content and advertisements were Instagram (36.7%) and YouTube (35.7%), respectively. Twitter (3.1%), Pinterest (7.0%), and Facebook (7.3%) were the least chosen mediums for viewing food content and advertisements.

### 3.4. Interaction with Food Content on Social Media

The related information is depicted in Table 3. About half of the study participants (51.6%) said that they follow food content creators on social media. In total, 54.5% of participants said that they feel like ordering food after watching an AD or content piece on social media. Almost half of the participants (50.3%) said that they feel so hungry that they order food right away. This trend was found to be significant (*p* = 0.009). About 28.3% of participants were always attracted to food on social media and 42% were sometimes attracted to food on social media. Many participants, (61.8%) loved viewing food content, while 38.2% did not like advertisements and food content appearing on their social media feeds. A significant association between love for watching food content and gender (boys) was observed (*p* = 0.009).

A significant association between liking food content posted by others and gender (girls) was found (*p* = 0.019). Some of the most followed food bloggers and content creators on social media according to the study were Bake with Shivesh, Kabita’s Kitchen, Chef Ranveer Brar, Natasha Gandhi, Saloni Kukreja, Nisha Madhulika, Cooking Shooking, LostandHungry, Ramenhairedgirl, Zingy Zest, So sauté, Priyanka Tiwari, Sanjyot Keer, and Meghna’s Food Magic.

### 3.5. Frequency of Food Advertisements Seen on Social Media

Figure 2 shows the frequency of advertisements seen on social media platforms by participants. In total, 30% of participants reported that they viewed food advertisements on social media every day; 11.6% of participants reported that food advertisements and content appear on their social media feeds after every few minutes, while 19.7% of participants reported that they viewed advertisements on social media after every few stories and posts.

### 3.6. Types of Food Advertisements Seen on Social Media

Table 4 presents information about the most seen AD categories on social media, and its association with BMI. Nearly 46.2% of participants said they viewed chocolate advertisements on social media regularly. More than half of the participants (64.3%) said they viewed advertisements by food delivery platforms and aggregators like Zomato and Swiggy regularly. About 35.4% of participants said they frequently viewed advertisements for ready-to-eat (RTE) foods like instant noodles, chips, pasta, namkeens, etc., regularly on social media. A significant association was also observed between viewing ADs of RTE foods and BMI (*p* = 0.004).

### 3.7. Association Between Social Media Exposure and Ordering Behaviour

Table 5 summarizes the association between exposure to food advertisements and ordering-related behaviours. The chi-square test showed no significant association between the frequency of food ADs viewed on social media and the tendency to order food after viewing an AD (*p* = 0.136). Similarly, no significant association was observed between ordering food after viewing ADs and gender (*p* = 0.425). However, participants who reported ordering food after watching ADs had significantly higher mean VAS scores compared to those who did not (2.38 ± 0.70 vs. 2.09 ± 0.73; *p* < 0.001), indicating greater overall food temptation. The ANOVA analysis showed no significant difference in VAS scores across age categories (*p* = 0.706).

A multivariable logistic regression model adjusting for sex, age, occupation, education and BMI showed that higher BMI (overweight/obese category) was significantly associated with higher odds of ordering food after viewing ADs (adjusted OR = 2.09, *p* = 0.043). More frequent exposure to food ADs showed a positive but borderline significant association with ordering behaviour (e.g., freq category 3: OR = 2.65, *p* = 0.054). In an adjusted linear regression model predicting VAS score, higher advertisement frequency remained significantly associated with higher VAS scores (*p* < 0.01), suggesting that frequent exposure to food content may increase overall food craving and temptation. These findings indicate a positive direction of association between food advertisement exposure, craving response, and ordering behaviour.

Multiple linear regression showed that exposure to food advertisements very frequently (categories 3 and 4) was significantly associated with higher VAS craving scores (*p* = 0.002 and *p* = 0.009, respectively), even after adjustment for demographic variables. These findings suggest that increased exposure to food advertisements and higher BMI are associated with a greater tendency to order food online and higher food craving levels.

## 4. Discussion

The study examined whether social media exposure and ADs of food content influence food choices among young adults belonging to Gen Z. The findings indicate that the intensity of digital food AD exposure was associated not only with food choices but also with higher BMI levels. The results of the study indicated that the participants had several unhealthy eating habits. Half of them, i.e., 50%, never eat breakfast, while 27.7% only sometimes eat breakfast, which is a matter of serious concern. This can be attributed to the influence of meal frequency and its timing. Studies have found that those who eat breakfast have a reduced risk of weight gain when compared to those who skip breakfast [24]. Evidence suggests those who eat breakfast as the biggest meal and ate only main meals and not snacks had a lower BMI [24]. In our sample, a considerable proportion of participants reported skipping breakfast; however, this behaviour was not incorporated as a variable in the regression analysis. It is therefore possible that skipping breakfast, snacking frequency, or irregular meal timing may have contributed to the association observed between social media food exposure and BMI or ordering behaviour. Future studies should statistically adjust for these eating-pattern variables to better determine their independent contribution. Our study findings show that most participants use social media more than 2 h a day. Our study also shows that 36.7% of participants used Instagram, and Instagram and YouTube are the most used platforms for viewing food content and advertisements. As of 2024, Instagram’s user base remains predominantly young, with 61.1% of users aged 18–34. Specifically, 30.8% are 18–24 years old, representing their core user base. Instagram continues to be a favourite among Gen Z, with 61% of users aged 16–24 using it weekly. Brands want to showcase themselves as authentic and creative on Instagram, and thus they use it for ADs and creating content. A possible explanation for these behaviours lies in the persuasive design strategies used in social media marketing. Visual cues such as close-up food images, descriptive sensory language, and influencer-driven endorsements may activate reward pathways and stimulate impulsive decision making, especially in digitally immersed youth [25,26].

Inferential analyses supported these findings. The pattern of findings suggests that young adults who are frequently exposed to appetitive food advertisements may experience heightened craving responses that drive impulsive food-ordering behaviour [27]. Rather than reflecting physiological hunger, these reactions appear to align with hedonic or reward-based eating triggered by visual cues [28]. Repeated exposure to highly esthetic food imagery, especially through personalized, algorithm-driven feeds, may reinforce these cravings over time, increasing susceptibility to impulse purchases [27,29]. The convergence of craving responses and ordering tendencies highlights a behavioural pathway through which digital food marketing may influence diet quality among Gen Z consumers. These results also add to previous research demonstrating the association between digital marketing and the buying behaviour of adolescents [30]. The study found that young adults watch a lot of food-related content on their social media in terms of advertisements as well as video and photography content by food agencies and independent creators. In our study, nearly 46.2% of participants said they viewed chocolate advertisements on social media regularly. A recent Australian analysis also reported that 55% of young adults encountered digital soft drink advertisements weekly and that increased exposure predicted higher intake of advertised foods [31]. This raises concerns given the frequency of food advertisements and the content young adults are constantly exposed to.

Our research study found that 95.2% of participants used social media during their free time, while 22.6% used it during mealtimes. Almost 23.8% of those using social media during mealtimes fell in the overweight, pre-obese, and obese categories. Our findings show that watching frequent ADs by food delivery platforms like Swiggy and Zomato is associated with BMI levels. We also observed that watching advertisements for RTE Foods like instant noodles, chips, chocolates, etc., was associated with higher BMI. A study also found that exposure to a particular advertisement for food or a specific drink on social media feeds is associated with a high intake of the same [31].

This study was the first of its kind at least in the state of Gujarat to evaluate the impact of digital marketing and social media content on the dietary choices of young adults. The study also produced detailed information about the type of social media platforms most used by study participants, the bloggers, and food creators they love viewing, and the kind of content that is most preferred by them.

A key strength of this study is its focus on Generation Z, a demographic under-represented in nutritional behaviour research despite being the highest consumer of social media. The use of a validated Visual Analogue Scale to assess food cue reactivity adds further robustness. Regression-based modelling provides deeper insights into associations beyond descriptive comparisons.

However, the limitations of this study include the cross-sectional nature of this study, which restricts the capacity to draw causal conclusions. In our study, all social media information was self-reported. Socioeconomic status and physical activity were not measured, and failure to adjust for these known determinants of weight and dietary behaviour may have resulted in residual confounding.

## 5. Conclusions

This study demonstrated an association between exposure to food-related social media content, and food-ordering behaviour and craving responses among young adults of Generation Z. Individuals who frequently engaged with appetitive food imagery and food delivery advertisements were more likely to report impulse-driven ordering and stronger craving responses, particularly those in higher BMI categories. However, due to the cross-sectional design, causality cannot be established, and it remains unclear whether social media exposure influences dietary behaviour or whether individuals with pre-existing behavioural tendencies are more responsive to digital cues.

The reliance on self-reported social media usage, and the absence of socioeconomic and physical activity variables must also be acknowledged. These limitations suggest that the findings should be interpreted cautiously.

Future research, employing longitudinal or experimental designs, is necessary to elucidate temporal relationships, clarify underlying mechanisms, and investigate psychological factors such as stress-induced or hedonic eating. Broader samples across diverse geographic and demographic groups will further strengthen the evidence base.

## Figures and Tables

**Figure 1 nutrients-17-03930-f001:**
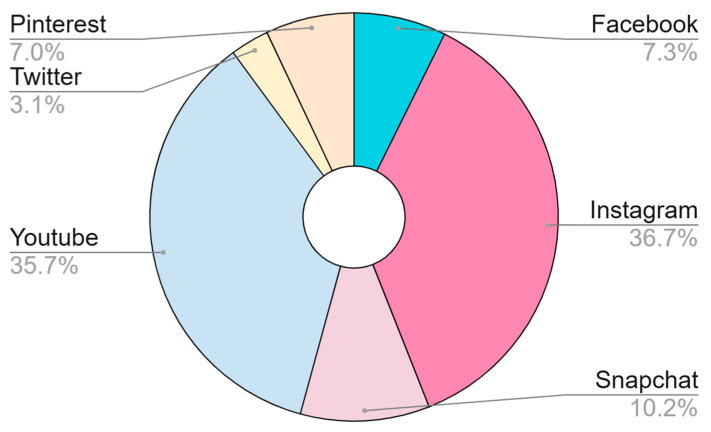
Platform of choice for watching/viewing food content and advertisements.

**Figure 2 nutrients-17-03930-f002:**
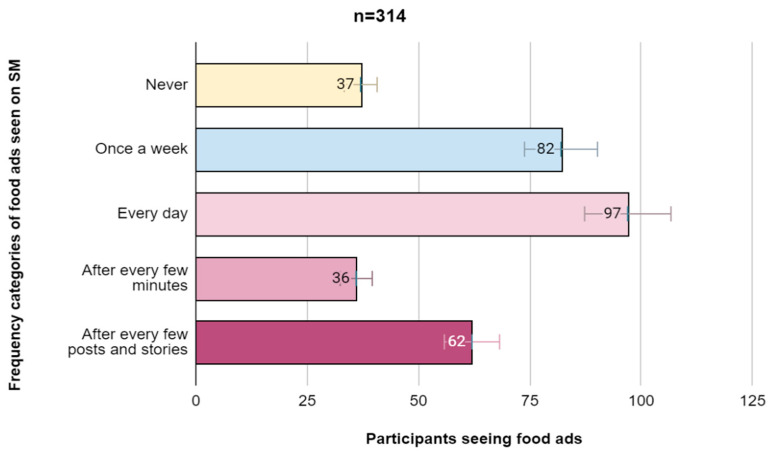
Frequency of advertisements seen on social media.

**Table 1 nutrients-17-03930-t001:** General characteristics of participants by gender.

Variable	Mean ± SD	*n* (%)	Boys (*n* = 146)	Girls (*n* = 168)	*p*-Value *
Age (In years)	19.7 ± 1.50	314	19.7 ± 1.48	19.69 ± 1.49	0.025 *
<20		162 (51.5%)	40 (27.3%)	62 (36.9%)	
≥20		152 (48.4%)	122 (83.5%)	106 (63%)	
Occupation	-	314			0.560
Student		287 (91.4%)	132 (90.4%)	155 (92.3%)	
Professional		27 (8.6%)	14 (9.6%)	13 (7.7%)	
Highest Educational Qualification	-	314			0.752
Higher Secondary		215 (68.5%)	103 (70.5%)	112 (66.7%)	
Bachelor’s Degree		94 (29.9%)	41 (28.1%)	53 (31.5%)	
Master’s Degree		5 (1.6%)	2 (1.4%)	3 (1.8%)	
BMI in kg/m^2^ (Asian cutoffs)	22.36 ± 4.41	314	23.52 ± 4.43	21.04 ± 4.44	0.000 *
Underweight (>18.5 kg/m^2^)		48 (15.3%)	11 (7.5%)	37 (22%)	
Normal (18.5–24.9 kg/m^2^)		140 (44.6%)	62 (42.5%)	78 (46.4%)	
Overweight/Probes/Obese (>25 kg/m^2^)		126 (40.1%)	73 (50%)	53 (31.5%)	
Disease History		146	168		-
No Disease history		300 (95.5%)	146	168	
Hyperthyroidism		1 (0.3%)	142 (97.3%)	158 (94%)	
Kidney Disease		2 (0.6%)	0 (0%)	1 (0.6%)	
Migraine		2 (0.6%)	2 (1.4%)	0 (0%)	
PCOS		8 (2.5%)	0 (0%)	8 (4.8%)	

* *p*-value for differences in frequencies by gender.

**Table 2 nutrients-17-03930-t002:** Eating patterns of participants.

Variable	Total *n* (%)	*p*-Value
Meals per day		
Three meals	153 (48.7%)	>0.05
More than three meals	73 (23.2%)	
Breakfast consumption		
Does not consume breakfast	157 (50%)	>0.05
Consumes breakfast sometimes	87 (27.7%)	
Hunger feelings		
Regularly feels hungry at mealtimes	136 (43.3%)	>0.05
Sometimes feels hungry other than meals	109 (34.7%)	

**Table 3 nutrients-17-03930-t003:** Variables depeicitng interaction with food content on social media.

Variable	*n*%	Boys (146)	Girls (168)	*p*-Value
Follow food content creators				0.452
Yes	162 (51.6%)	74 (50.7%)	78 (46.4%)	
No	152(48.4%)	72 (49.3%)	90 (53.6%)	
Feel like ordering food after watching AD/content				0.425
Yes	171 (54.5%)	70 (47.9%)	73 (43.5%)	
No	143 (45.5%)	76 (52.1%)	95 (56.5%)	
Feeling hungry and ordering right away				0.009 *
Yes	158 (50.3%)	84 (57.5%)	72 (42.9%)	
No	156 (49.7%)	62 (42.5%)	96 (57.1%)	
Attracted to food on social media				0.546
Yes	89 (28.3%)	47 (32.2%)	46 (27.4%)	
No	93 (29.7%)	42 (28.8%)	77 (28%)	
Sometimes	132 (42.0%)	57 (39.0%)	75(44.6%)	
Love food advertisements				0.009 *
Yes	194 (61.8%)	67 (45.9%)	53 (31%)	
No	120 (38.2%)	79 (54.1%)	115 (68.5%)	
Like viewing food advertisements/pictures posted by others				0.019 *
No	76 (24.2%)	45 (30.8%)	31 (18.5%)	
Yes	131 (41.7%)	51 (34.9%)	80 (47.6%)	
Sometimes	107 (34.1%)	50 (34.2%)	57 (33.9%)	
Food content creators/bloggers followed on social media				0.185
None of the above	139 (44.3%)	72 (49.3%)	67 (39.9%)	
Food bloggers and influencers	89 (28.3%)	38 (26%)	52 (31%)	
Chefs	52 (16.6%)	14 (9.6%)	11 (6.5%)	
All types of food creators	180 (57.3%)	22 (15.1%)	38 (22.6%)	

* *p*-value for differences in frequencies by gender.

**Table 4 nutrients-17-03930-t004:** Type of AD content viewed on social media by BMI.

Variable	*n* (%)	Underweight	Normal	Overweight/Obese	*p*-Value
Chocolate					0.266
No	169 (53.8%)	31 (64.6%)	73 (52.1%)	65 (51.6%)	
Yes	145 (46.2%)	17 (35.4%)	67 (47.9%)	61 (48.4%)	
Fast Food chain					0.063
No	112 (35.7%)	24 (50%)	49 (35%)	39 (31%)	
Yes	202 (64.3%)	25 (50%)	91 (65%)	87 (69%)	
Food delivery platforms					0.004 *
No	102 (32.5%)	27 (56.2%)	41 (29.3%)	34 (27%)	
Yes	212 (65.5%)	21 (43.8%)	99 (70.7%)	92 (73%)	
RTE Foods					0.004 *
No	203 (64.6%)	41 (85.4%)	87 (62.1%)	75 (59.5%)	
Yes	111 (35.4%)	79 (14.6%)	53 (37.1%)	51 (40.5%)	

* *p*-value for differences in frequencies by BMI. Abbreviations: AD, advertisement; RTE, ready-to-eat.

**Table 5 nutrients-17-03930-t005:** Summary of associations analyzed.

Predictor Variable	Statistical Tests	Result	*p*-Value	Interpretation
Frequency of AD × Ordering after AD	Chi-square	χ^2^ = 6.99	0.136	Not significant
Sex × Ordering after AD	Chi-square	χ^2^ = 0.63	0.425	Not significant
VAS craving score (ordering vs. not ordering)	Independent *t*-test	Difference = 0.30	<0.001	Ordering group had higher cravings
VAS × Age category	One way ANOVA	F = 0.14	0.706	No difference across age
BMI category: Ordering food after AD	Logistic regression	BMI Category 3 OR = 2.09	0.043	Higher BMIs are more likely to order after ADs

## Data Availability

The raw data supporting the conclusions of this article will be made available by the authors on request.

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
