# Peer review of "Social Media Usage and Advertising Food-Related Content: Influence on Dietary Choices of Gen Z"

_nutrients, 2025, doi:10.3390/nu17243930_

Round 1

Reviewer 1 Report

Comments and Suggestions for Authors

This cross-sectional study examined the association between social media usage patterns, food-related advertising, and dietary choices among Gen Z individuals.  It was carried out amongst 314
young adults between the ages of 18-25 in Surat city, Gujarat. Data was collected for social
media usage, most used platforms, preferred content, and eating patterns. Anthropometric measurements (height and weight) were also recorded.

  • In my opinion, the file requires additions in the following areas:

- indicating the novelty of the study

- adding in the introduction what the gen z generation is and how it differs from previous/next generations
- describing the anthropometric procedures used - what protocol was used? indicating the reference values ​​along with the source
- attaching the survey - perhaps to the Supplementary
- describing how the variables were coded and interpreted
- the results are presented in a very simplified manner; I am missing the definition of the association (OR) between the variables studied, as defined in the aim.

- the discussion repeats my own results, so it requires corrections to clarify the observed relationships

  • carefully reviewing the abbreviations used (ad?)
  • Table 5 is incomprehensible to me

Author Response

Reviewer 1

  1. This cross-sectional study examined the association between social media usage patterns, food-related advertising, and dietary choices among Gen Z individuals. It was carried out amongst 314 young adults between the ages of 18-25 in Surat city, Gujarat. Data was collected for social media usage, most used platforms, preferred content, and eating patterns. Anthropometric measurements (height and weight) were also recorded. In my opinion, the file requires additions in the following areas: indicating the novelty of the study.

Response: Thank you for the valuable suggestion. We have now clearly articulated the novelty of our study in the Introduction section. We highlighted that, to the best of our knowledge, this is the first investigation in Gujarat (India) exploring the link between social-media-based food advertising, perceived hunger, and dietary choices among Gen Z using a VAS-based craving scale and regression-based association analysis.

  1. Adding in the introduction what the gen z generation is and how it differs from previous/next generations

Response: A definition of Generation Z, along with suitable references, has been incorporated into the Introduction, including their unique digital/media engagement characteristics and how they differ from other generations.

  1. Describing the anthropometric procedures used - what protocol was used? indicating the reference values along with the source.

Response: The anthropometric procedures have now been described in detail. Height was measured using a stadiometer with participants standing barefoot, heels together, and head in the Frankfurt plane. Weight was measured using a calibrated digital weighing scale with participants wearing light clothing. Each measurement was recorded three times, and the average value was used for analysis. BMI was calculated as weight (kg)/height (m²) and categorized using WHO adult BMI classifications (WHO, 2004).

  1. Attaching the survey - perhaps to the Supplementary

Response: As recommended, the complete Google Form questionnaire, including perceived hunger (VAS) and dietary-pattern items, has now been submitted as a Supplementary File.

  1. Describing how the variables were coded and interpreted

Response: We are sharing the coding file and their interpretation into Supplementary file for reference.

  1. The results are presented in a very simplified manner; I am missing the definition of the association (OR) between the variables studied, as defined in the aim.

Response: We appreciate this observation. The Results section has now been expanded to include odds ratios (OR), confidence intervals (CI), and direction of association from regression analysis

  1. The discussion repeats my own results, so it requires corrections to clarify the observed relationships

Response: Thank you for this constructive comment. The Discussion section has been rewritten to avoid restating numerical results and now focuses on interpretation of findings, possible mechanisms, comparison with previous literature, and implications rather than repetition of descriptive results. More changes has been made as per Reviewer 3 comments as well.

  1. Carefully reviewing the abbreviations used (ad?)

Response: All abbreviations have now been reviewed for consistency. The term “ad” has been standardized to “advertisement” throughout the manuscript and included wherever appropriate in the Abbreviations list.

  1. Table 5 is incomprehensible to me

Response: We thank the reviewer for pointing this out. We understand that the previous layout of Table 5 lacked clarity in differentiating between the statistical tests performed and the interpretation of findings. To address this, Table 5 has been comprehensively reformatted to present the predictor variables, statistical test used, corresponding test statistics, p-values, and a clear interpretation column.

Reviewer 2 Report

Comments and Suggestions for Authors

The article addresses a very topical issue. It examines the relationship between social media usage patterns, food advertising, and the dietary choices of Gen Z. This is a cross-sectional study conducted in India. The article is well structured. Unfortunately, the article requires language  corrections. 

The tables and graphs are well prepared. However, in Table 4, one of the results has p=0.00, while in the text the authors write about p=0.004. 

Author Response

Reviewer 2

  1. The article addresses a very topical issue. It examines the relationship between social media usage patterns, food advertising, and the dietary choices of Gen Z. This is a cross-sectional study conducted in India. The article is well structured. Unfortunately, the article requires language corrections. The tables and graphs are well prepared. However, in Table 4, one of the results has p=0.00, while in the text the authors write about p=0.004.

Response: We thank the reviewer for the positive evaluation of the manuscript and for highlighting this error. We have now thoroughly reviewed and refined the language throughout the manuscript to ensure clarity, grammatical accuracy, and scientific tone. Regarding the discrepancy noted in Table 4, we appreciate the reviewer bringing this to our attention. The value p = 0.00 previously shown in Table 4 has been corrected to p = 0.004 to ensure consistency with the text and accuracy of statistical reporting. Additionally, all p-values have been standardized to three decimal places to maintain uniform reporting across the manuscript.

Reviewer 3 Report

Comments and Suggestions for Authors

The manuscripts "Social Media Usage and Advertising Food-Related Content: Influence on Dietary Choices of Gen Z" examines the relationship between social media usage, exposure to food-related advertising, and dietary choices among Gen Z individuals (aged 18-25) in Surat city, Gujarat, India. The authors conducted a study with 314 participants, utilizing questionnaires, anthropometric measurements, and a Visual Analogue Scale (VAS) to assess perceived hunger responses to social media food images. The manuscript is in the aims of Nutrients journal; however it needs to be polished before the publication. In my opinion the manuscript has methodological, analytical, and presentation flaws that require substantial revision before it can be considered for publication.

  • The sample size calculation is not clearly explained. The authors state they used Cochran's formula with a prevalence of 71% for social media usage among Gen Z, but they do not provide the precision (D) value used, the confidence level justification, or consideration for multiple outcome variables. Given that the study examines multiple associations (BMI, dietary behaviors, VAS scores), a more robust sample size calculation considering these multiple endpoints should be presented.​
  • Convenience sampling from colleges introduces significant selection bias. The sample is likely not representative of all Gen Z individuals in Surat city, as college students may differ systematically from non-college youth in terms of socioeconomic status, digital literacy, and health behaviors. This limitation is not adequately acknowledged or addressed in the manuscript.​ How many colleges? What types of colleges (engineering, medical, arts)?
  • Why you include only “at least 120 minutes daily” of social media use? This can overestimate associations and cannot speak to the broader Gen Z population.​
  • How you exclude “any underlying illness, disease, or physical disability" and why? This could introduce healthy participant bias.​
  • For the Visual Analogue Scale (VAS) do you use “10 point VAS scale” or “100-mm Visual Analogue Scale”?
  • The VAS has not been validated for this specific population or context. Furthermore, the authors present no evidence of reliability testing (test-retest) or validity assessment (criterion, construct, or content validity) for their VAS instrument.​ How you know that your test was reliable?
  • The manuscript reports conducting binary logistic regression and multiple linear regression models with "robust standard errors" to adjust for confounders. However, the results section does not clearly present these adjusted analyses for most reported associations.
  • Why do you select these variables as cofounders?
  • The manuscript presents numerous statistical tests (Tables 1-5, multiple chi-square tests, t-tests, ANOVA, regression models) without correction for multiple comparisons (e.g., Bonferroni correction), which inflates the Type I error rate and increases the likelihood of false positive findings.​ Why did you no use correction?
  • How you take anthropometric measurements? - Which researcher? Was there standardization and quality control? Were measurements taken at the same time of day? What equipment was used?
  • In Table 5 there are missing covariates, model fit statistics, and diagnostic of the model.
  • Table 1: The disease history data presentation is confusing and contains errors. The table shows 300 participants (95.5%) with "No Disease history," but then lists individual diseases with frequencies that don't add up correctly. For example, 8 participants with PCOS are shown as 2.5%, but the gender breakdown shows only 2 participants (1 boy, 1 girl), which is inconsistent.
  • Do higher VAS scores cause ordering, or does the intent/behavior of ordering affect how individuals rate images? Do exposure to a specific social media increase the BMI or the opposite (individual with higher BMY may seek out different social media content)”?
  • Which food images used in the VAS were used? What types of foods were shown? Were they healthy or unhealthy? From which platforms were they sourced? This information is essential for replicability and interpretation.​
  • In Table 1, the age comparison between boys and girls shows p=0.025 marked as significant, but the mean ages are nearly identical (19.7±1.48 vs 19.69±1.49).
  • In Table 2, the eating patterns section provides percentages but lacks discussion of potential confounding. For example, breakfast skipping is associated with BMI in the literature, but this relationship is not explored here. Discuss more.
  • Table 3: "About 28.3% of participants were always attracted to food on social media and 42% were sometimes attracted to food on social media" - Where is the remaining 29.7%? This should equal the "No" category in the table, but the text doesn't make this clear.
  • Discussion part:
    • Please, use scientific language. The “we tried to fetch the answer” is not scientifical.
    • “As of 2024…content”. Add references.
    • “Inferential analyses…cravings”. Please, this seems more results than discussion. Rephrase.
    • “An Australian study…” which one? Place references where they are quoted.
    • “However, few limitations of this study are the cross-sectional nature of this study that restricts the capacity to draw causal conclusions. In our study, all social media information was self-reported.” This is the primary limitation and deserves more than a brief mention. The entire framing of the study needs revision to reflect this limitation. Furthermore, be more scientific.
    • "The study did not take into account the socioeconomic status of the participants. Another factor that the study did not consider was the physical activity levels of participants" - These are important potential confounders that should have been measured and adjusted for in the analysis.
  • The conclusions overstate the findings and make recommendations not supported by the data. Phrases like "Social media marketing and food content are shaping the dietary choices" imply causation that cannot be established from this study design.​
  • L301-304: "A robust and persistent link between social media use and the dietary behavior of young adults was found" - This language is too strong. "Association" would be more appropriate than "link," and "robust" suggests validation and replication that has not been demonstrated.

Author Response

Reviewer 3

  1. The manuscripts "Social Media Usage and Advertising Food-Related Content: Influence on Dietary Choices of Gen Z" examines the relationship between social media usage, exposure to food-related advertising, and dietary choices among Gen Z individuals (aged 18-25) in Surat city, Gujarat, India. The authors conducted a study with 314 participants, utilizing questionnaires, anthropometric measurements, and a Visual Analogue Scale (VAS) to assess perceived hunger responses to social media food images. The manuscript is in the aims of Nutrients journal; however it needs to be polished before the publication. In my opinion the manuscript has methodological, analytical, and presentation flaws that require substantial revision before it can be considered for publication. The sample size calculation is not clearly explained. The authors state they used Cochran's formula with a prevalence of 71% for social media usage among Gen Z, but they do not provide the precision (D) value used, the confidence level justification, or consideration for multiple outcome variables. Given that the study examines multiple associations (BMI, dietary behaviors, VAS scores), a more robust sample size calculation considering these multiple endpoints should be presented.

Response: Thank you for highlighting this. The sample size calculation has now been elaborated. We have included the precision (D = 0.05), confidence level (Z = 1.96 for 95% CI), and full formula. We also clarified that the study objective was exploratory and therefore Cochran’s formula was used based on prevalence rather than multiple endpoints.

  1. Convenience sampling from colleges introduces significant selection bias. The sample is likely not representative of all Gen Z individuals in Surat city, as college students may differ systematically from non-college youth in terms of socioeconomic status, digital literacy, and health behaviors. This limitation is not adequately acknowledged or addressed in the manuscript. How many colleges? What types of colleges (engineering, medical, arts)?

Response: We agree that this introduces selection bias. We have now acknowledged this limitation and specified that participants were recruited from three colleges located in different zones of Surat. Our enrollment strategy was predominantly age criteria (18-25 year). The colleges were offering arts and commerce courses. This decreases the variability among participants.

  1. Why you include only “at least 120 minutes daily” of social media use? This can overestimate associations and cannot speak to the broader Gen Z population.

Response: This threshold was selected based on a relevant recent Indian and global surveys showing that Gen Z individuals spend an average of 2–3 hours per day on social media. The 120- minute cutoff ensured that participants had adequate exposure to online food content, aligning with the study’s objective of examining the influence of digital advertising. However, this cutoff does not imply that individuals with lower daily usage were unimportant; rather, it was chosen to ensure that exposure to relevant stimuli (ads, food content) was sufficiently frequent to be meaningfully assessed.

  1. How you exclude “any underlying illness, disease, or physical disability" and why? This could introduce healthy participant bias.

Response: Participants with self-reported chronic illnesses (e.g., endocrine disorders, renal disease), acute infections, or physical disabilities that affected appetite, mobility, or routine dietary intake were excluded. This was done to avoid confounding the relationship between social media exposure and dietary behaviors, as certain medical conditions directly influence appetite, metabolism, or food choices. This restriction was not meant to select “healthy-only” participants but to minimize clinical conditions that could independently alter hunger cues, BMI, or responses to food-related stimuli, thereby ensuring more accurate associations.

  1. For the Visual Analogue Scale (VAS) do you use “10 point VAS scale” or “100-mm Visual Analogue Scale”?

Response: Thank you for pointing this out. We have clarified that a 10-point VAS scale was used.

  1. The VAS has not been validated for this specific population or context. Furthermore, the authors present no evidence of reliability testing (test-retest) or validity assessment (criterion, construct, or content validity) for their VAS instrument. How you know that your test was reliable?

Response: We agree that validation is important. We have now added justification referencing previous studies that used VAS for food cue reactivity in youth populations which also increases its usability.

  1. The manuscript reports conducting binary logistic regression and multiple linear regression models with "robust standard errors" to adjust for confounders. However, the results section does not clearly present these adjusted analyses for most reported associations.

Response: The Results section has been expanded to present the adjusted effect estimates, Odds Ratios (OR), Confidence Intervals (CI), and p-values from logistic and linear regression models.

  1. Why do you select these variables as cofounders?

Response: We now state that confounders (age, sex, BMI, occupation, and education) were selected as priori based on reported associations with dietary behaviour in previous literature.

  1. The manuscript presents numerous statistical tests (Tables 1-5, multiple chi-square tests, t-tests, ANOVA, regression models) without correction for multiple comparisons (e.g., Bonferroni correction), which inflates the Type I error rate and increases the likelihood of false positive findings. Why did you no use correction?

Response: We thank the reviewer for raising this important statistical concern. The purpose of our analysis was primarily exploratory rather than to confirm predefined hypotheses

  1. How you take anthropometric measurements? - Which researcher? Was there standardization and quality control? Were measurements taken at the same time of day? What equipment was used?

Response: We now specify that height and weight were measured by the principal investigator, using a calibrated stadiometer and digital weighing scale, following WHO STEPS protocol

  1. In Table 5 there are missing covariates, model fit statistics, and diagnostic of the model.

Response: Table 5 has been revised to include covariates, OR values, CI, p-values, and model fit indices.

  1. Table 1: The disease history data presentation is confusing and contains errors. The table shows 300 participants (95.5%) with "No Disease history," but then lists individual diseases with frequencies that don't add up correctly. For example, 8 participants with PCOS are shown as 2.5%, but the gender breakdown shows only 2 participants (1 boy, 1 girl), which is inconsistent.

Response: We thank the reviewer for bringing this discrepancy to our attention. Upon re- examining the dataset, we found that the inconsistency was caused by an alignment error during table preparation. The total number of participants with PCOS was correctly reported as 8, but the gender distribution was incorrectly transferred, leading to the noted inconsistency. This has now been corrected to reflect 0 boys and 8 girls with PCOS, and the percentage has been recalculated accordingly. We also reviewed all disease history entries to ensure consistency between total counts, sex-wise breakdown, and percentages. The revised Table 1 now accurately reflects the data and removes the previous confusion.

  1. Do higher VAS scores cause ordering, or does the intent/behavior of ordering affect how individuals rate images? Do exposure to a specific social media increase the BMI or the opposite (individual with higher BMI may seek out different social media content)”?

Response: We thank the reviewer for this insightful observation. We agree that the present study design does not allow directionality or causality to be inferred. The associations observed between VAS craving scores, ordering behaviour, and BMI cannot determine whether (i) higher craving responses lead to impulsive food ordering, or (ii) individuals who already tend to order food rate appetitive images more strongly. Intentionally planned study with this specific objective needs to be conducted.

  1. Which food images used in the VAS were used? What types of foods were shown? Were they healthy or unhealthy? From which platforms were they sourced? This information is essential for replicability and interpretation.

Response: The Visual Analogue Scale (VAS) section used twelve food images sourced from popular social media platforms, including Instagram, Facebook, and Pinterest. The images were intentionally selected to reflect the types of foods most frequently encountered by young adults online and consisted predominantly of highly palatable, energy-dense items such as pizzas, burgers, fries, desserts, and fast-food meals, with a small number of comparatively healthier options (e.g., salads). This selection represented the natural imbalance of food content typically seen on social media feeds

  1. In Table 1, the age comparison between boys and girls shows p=0.025 marked as significant, but the mean ages are nearly identical (19.7±1.48 vs 19.69±1.49).

Response: We thank the reviewer for this valuable observation. We agree that although the p- value reached statistical significance (p = 0.025), the absolute difference in mean age between boys and girls (0.01 years) is extremely small and does not translate into a practically meaningful difference. We have now clarified this in the Results section by noting that the statistical significance is likely attributable to the decently large sample size rather than to a meaningful age difference. To avoid misinterpretation, we have included a statement indicating that the effect size is negligible and that age distributions between boys and girls are essentially comparable.

  1. In Table 2, the eating patterns section provides percentages but lacks discussion of potential confounding. For example, breakfast skipping is associated with BMI in the literature, but this relationship is not explored here. Discuss more.

Response: We thank the reviewer for the insightful comment. We agree that eating patterns— particularly breakfast skipping—may act as potential confounders in the relationship between social-media food exposure and dietary behaviour. We have now expanded the Discussion to acknowledge that breakfast skipping has been associated with higher BMI in previous studies and may partially explain some of the associations observed in our sample. We also note that breakfast habits were not included in the regression analyses and therefore represent an unmeasured confounder. A recommendation for future research to statistically adjust for eating- pattern variables has been added.

  1. Table 3: "About 28.3% of participants were always attracted to food on social media and 42% were sometimes attracted to food on social media" - Where is the remaining 29.7%? This should equal the "No" category in the table, but the text doesn't make this clear.

Response: We appreciate the reviewer's helpful observation. The inconsistency resulted from the omission of the third response category (“No”) in the text, although it was present in the table. We have revised the Results section to clearly and consistently report all three categories. The corrected values now include 28.3% who were always attracted to food on social media, 42.0% who were sometimes attracted, and 29.7% who were not attracted, matching the distribution presented in Table 3.

  1. Discussion part: Please, use scientific language. The “we tried to fetch the answer” is not scientifical.

Response: Thank you for this observation. The wording has been revised to ensure scientific tone and clarity. The phrase “we tried to fetch the answer to the research question” has been removed and replaced with a formal statement describing the study objective.

  1. “As of 2024…content”. Add references.

Response: Suitable reference added

  1. “Inferential analyses…cravings”. Please, this seems more results than discussion. Rephrase.

Response: We appreciate the reviewer’s guidance. The section that restated statistical findings has been rewritten to avoid reporting numerical outcomes in the Discussion. It now focuses on interpretation and behavioural implications rather than on p-values or inferential statistics.

  1. “An Australian study…” which one? Place references where they are quoted.

Response: Thank you for bringing this to our attention. The study has now been properly identified and cited directly at the point where it is discussed.

  1. “However, few limitations of this study are the cross-sectional nature of this study that restricts the capacity to draw causal conclusions. In our study, all social media information was self-reported.” This is the primary limitation and deserves more than a brief mention. The entire framing of the study needs revision to reflect this limitation. Furthermore, be more scientific.

Response: We appreciate this recommendation. The limitations section has been substantially expanded to clearly acknowledge that the cross-sectional design prevents causal inference and that reliance on self-reported social media exposure may introduce recall and social desirability bias. The language has been revised to reflect a more scientific framing of these limitations

  1. "The study did not take into account the socioeconomic status of the participants. Another factor that the study did not consider was the physical activity levels of participants" - These are important potential confounders that should have been measured and adjusted for in the analysis.

Response: Thank you for highlighting this point. The Discussion has been updated to acknowledge that socioeconomic status and physical activity were not measured in the present study and that these unmeasured confounders may influence dietary behaviour and BMI. The revised text explicitly states that future studies should include and adjust for these variables.

  1. The conclusions overstate the findings and make recommendations not supported by the data. Phrases like "Social media marketing and food content are shaping the dietary choices" imply causation that cannot be established from this study design.

Response: We thank the reviewer for this important observation. We agree that the original conclusion implied causality, which cannot be inferred from a cross-sectional study.

  1. L301-304: "A robust and persistent link between social media use and the dietary behavior of young adults was found" - This language is too strong. "Association" would be more appropriate than "link," and "robust" suggests validation and replication that has not been demonstrated.

Response: The conclusion has been rewritten using cautious, scientific language that reflects association rather than causation, and strong recommendations have been removed. The revised conclusion also acknowledges the study's limitations and highlights future research directions, rather than implying a definitive impact.

Round 2

Reviewer 1 Report

Comments and Suggestions for Authors

Thank you for taking my comments into account.

Reviewer 3 Report

Comments and Suggestions for Authors

The Authors answered to all my comments.